# The Secrets of the Mediterranean Diet. Does [Only] Olive Oil Matter?

**DOI:** 10.3390/nu11122941

**Published:** 2019-12-03

**Authors:** Alessandra Mazzocchi, Ludovica Leone, Carlo Agostoni, Isabella Pali-Schöll

**Affiliations:** 1Department of Clinical Sciences and Community Health, University of Milan, 20126 Milan, Italy; alessandra.mazzocchi@unimi.it (A.M.); ludovica.leone@unimi.it (L.L.); 2SIGENP (Italian Society of Pediatric Gastroenterology, Hepatology, and Nutrition), via Libero Temolo 4 (Torre U8), 20126 Milan, Italy; 3Pediatric Intermediate Care Unit, Fondazione IRCCS Ca’ Granda Ospedale Maggiore Policlinico, 20126 Milan, Italy; 4Center for Pathophysiology, Infectiology and Immunology, Institute of Pathophysiology and Allergy Research, Medical University of Vienna, 1090 Vienna, Austria; isabella.pali@vetmeduni.ac.at; 5Comparative Medicine, The interuniversity Messerli Research Institute, University of Veterinary Medicine, Medical University and University Vienna, 1090 Vienna, Austria

**Keywords:** allergy, cancer, CHD/CVD, fresh food, inflammation, metabolic syndrome, mono-unsaturated fatty acids, obesity, olive oil, traditional

## Abstract

Diet plays a key role in the maintenance and optimal functioning of immune cells. The Mediterranean dietary pattern is an example of a prudent choice of lifestyle and scientifically accepted to help preserve human health by protecting against major chronic and inflammatory diseases. Mediterranean diets (MedDiets) are characteristically high in the consumption of fruits, vegetables and salad, bread and whole grain cereals, potatoes, legumes/beans, nuts, and seeds. Their common central feature is the usage of olive oil as the main source of fat. The health benefits attributed to olive oil are specifically related to extra virgin olive oil (EVOO) intake with its high nutritional quality and multiple positive effects on health. Overall, MedDiets have direct (mono-unsaturated fatty acids (MUFAs), tocopherols, polyphenols) and indirect (low saturated fats, well-balanced linoleic/alpha linolenic acid) effects on the immune system and inflammatory responses. In the present paper, we summarize the current knowledge on the effect of olive oil per se and MedDiets generally on immune-mediated and inflammatory diseases, such as coronary heart disease (CHD)/cardiovascular diseases (CVD), obesity, type-2 diabetes, cancer, asthma, and allergies.

## 1. Introduction

The term immunomodulation refers to any therapeutic intervention directed at modifying the immune response. At birth, the immune system is immature, but it develops with age, antigen stimulation, and appropriate nutrition [1]. Diet plays a key role in the maintenance and optimal functioning of immune cells. It is well known that nutrients and nutritional factors can help to preserve good health, influencing all aspects of human biology by connecting nutrient metabolism, gut microbiota, and immune system [2].

For instance, essential fatty acids (EFAs) are important immune regulators. Linoleic acid (LA), the parental *n*-6 polyunsaturated fatty acid (PUFA), is converted in the mammalian body into arachidonic acid (AA) and subsequently may give origin to pro-inflammatory lipid mediators (eicosanoids). On the contrary, from α-linolenic acid (ALA), *n*-3 PUFA eicosapentaenoic acid (EPA) and docosahexaenoic acid (DHA) are produced and subsequently converted into anti-inflammatory and/or pro-resolving lipid mediators (resolvins and protectins) [3].

The prevalence in the membranes of inflammatory cells of either AA, or EPA/DHA, may influence human immunologic conditions, and since *n*-3 and *n*-6 PUFAs compete for the same metabolic pathways, an increase of *n*-3 PUFA rather than *n*-6 PUFA will enhance an anti-inflammatory response. 

Vitamins and trace elements are also involved in the maintenance of the immune response. The Mediterranean dietary pattern is not only exemplified as a prudent choice of lifestyle, but also as a scientifically accepted mechanism that helps to preserve human health and protect against major chronic diseases. Mediterranean-style diets are considered beneficial for the prevention and/or treatment of obesity, type-2 diabetes, and inflammatory disorders [4]. 

In the present paper, we summarize the current knowledge on the effect of the Mediterranean diet (MedDiet) with special focus on extra virgin olive oil (EVOO) on immune-mediated inflammatory diseases (IMID) and cardiovascular diseases (CVD).

## 2. Definition and Composition of Mediterranean Diet

The term “Mediterranean diet” (MedDiet) refers to the pattern of the diet that was common in the 1960s in the olive tree-growing countries around the Mediterranean Sea, especially Greece/Crete and the south of Italy. At that time, the Seven Countries Study showed that mortality due to coronary heart diseases in the Mediterranean area was 2–3 times lower than in North Europe and the USA [5]. The time point is important to consider, because in many of these countries the diet has changed since then and is now similar to the so-called “Western diet” of the USA and Northern Europe.

More than 16 countries border the Mediterranean Sea, the inhabitants of which follow a different dietary pattern according to cultural, social, or religious habits. However, there is an overall Mediterranean dietary pattern (comprehensively reviewed in [6]), which is characterized by the high consumption of fruits, vegetables and salad, bread and whole grain cereals, potatoes, legumes/beans, nuts, and seeds. Dairy products, like cheese and yogurt, fish, shellfish, and poultry are consumed in low to moderate amounts, whereas little red and processed meat is eaten, and eggs are consumed up to four times a week. The need for salt and fat for aromatic purposes is lowered by wide usage of herbs and spices. Wine and/or other fermented beverages are consumed in low to moderate amounts, accompanying the meals.

Finally, a common and central feature of all variations of the MedDiet is the usage of olive oil as the main source of fat, with mainly monounsaturated fatty acids.

Altogether, this diet is low in saturated fatty acids and provides high amounts of antioxidants, carbohydrates, and fiber and, above all, a high content of monounsaturated fatty acids and *n*-3 PUFAs, mainly derived from olive oil as alpha-linolenic acid and long-chain PUFAs from fish, particularly in certain areas.

Apart from the food groups consumed, food is produced traditionally and marketed locally, and is mostly used and eaten shortly after harvesting. Eating in a pleasant, familiar environment, having a rest after the meal and also being physically and socially active completes the dietary pattern to a healthy lifestyle (Figure 1). Recent papers also provide evidence that the MedDiet—beyond its benefits for human health—has a low adverse effect on the environment and represents an affordable model of sustainability [7]. 

The composition of a MedDiet can also be broken down to the actual macro- and micronutrient content (Table 1). It was shown that the content in MedDiet was nearly totally according to the actual guidelines and recommendations for daily nutrient uptake even for adults living in the US (as given in Table 1). Taking a closer look into the composition of a MedDiet, and what was especially seen in Greece [8], the resulting *n*-6: *n*-3 ratio is 2:1 compared to a ratio of 15:1 and higher in Western and Northern Europe or the USA, where currently a ratio of 5:1 is recommended. In addition to the desired fatty acids (FA) composition, the Greek and Cretan non-fat components include considerable amounts of antioxidants, phytochemicals, and phenolic compounds, together with vitamin C and E, β-carotene as precursor of vitamin A, glutathione, resveratrol, selenium, phytoestrogens, and folate. The main dietary sources providing these valuable compounds are olive oil, wine, fish, and vegetables, especially tomatoes, onions, garlic, and herbs like oregano, mint, rosemary, parsley, and dill [8].

## 3. Mediterranean Diet and Diet Diversity

An additional measure to consider when defining the value of a certain nutritional pattern—for health, growth, and immunological events—is diet diversity [12]. We have summarized the influence of diet diversity on asthma and allergy development in a position paper [13], where the definition of diet diversity is “...the number of different foods or food groups consumed over a given reference time period”. Such an independent measure based on desirable dietary patterns is the Mediterranean Diet Score [14], suggested by Trichopoulou et al. in 2003 [15]. Here, nine components are considered, which are vegetables, legumes, fruits and nuts, cereals, fish and seafood, ratio of mono-unsaturated fatty acids (MUFA):SFA, meat and meat products, dairy products, and moderate alcohol consumption. A point is scored when a person takes in more than the median of the same sex group from the first six components listed above (which are according to MedDiet), another point when taking in lower than the median from meat and dairy products (not consistent with MedDiet), and another when alcohol intake is moderate (which is max 25 g/day for female and 50 g/day for male individuals). Therefore, the maximum possible points are nine, and the diversity score can range from zero to nine. Several variations of the MedDiet Score have been created to be applied in studies with different populations, also outside the Mediterranean area, providing the possibility to measure adherence to MedDiet [6], like the alternate MedDiet score [16], the 14-Point MedDiet Adherence Screener (MEDAS) [17], the MedDiet 55 Score (MD55) [18], and the Med Adequacy Index (MAI) [19]. These indices all vary for which foods are considered as being in one group (e.g., nuts and legumes together, all kinds of meat within one group or just red meat, and separately considering usage of olive oil). In summary, the MedDiet reflects a healthy eating pattern, which has been found to protect against several inflammatory diseases like coronary heart disease (CHD) and stroke [16].

## 4. Mediterranean Diet Contribution to Immunomodulation

### 4.1. Bioactive Components

Olive oil is the traditional symbol of the Mediterranean diet, representing the primary source of fat. The health benefits attributed to the consumption of olive oil are specifically related to extra virgin olive oil (EVOO), which is considered a key bioactive food because of its high nutritional quality (Table 2) [20]. 

Traditionally, the high content of MUFAs was considered to be responsible for the protective effects of EVOO. Indeed, the concentration of the MUFA oleic acid (C18:1 *n*-9) is much higher (55–83%) than that of the other fatty acids (linoleic, palmitic, or stearic acids), which range between 3% and 21%. However, now it is known that most of these benefits are related to the minor components in the unsaponifiable fraction (about 2% of oil weight), including phenolic compounds, phytosterols, tocopherols, and pigments. This fraction is responsible for EVOO oxidative stability, sensorial attributes (such as bitterness and pungency) and its unique fragrance [21]. The presence of these molecules depends on the grade of maturation of the fruit, the cultivar variety, the climate of the area of cultivation, and the type of oil extraction processes [22]. Phenolic concentration in EVOO range from 50 to 800 mg/kg [23]. The phenolic content consists of various phenolic classes, such as phenolic acids (vanillic, coumaric, caffeic, protocatechuic, p-hydroxybenzoic, ferulic), lignans (acetoxypinoresinol, pinoresinol), flavones (apigenin, luteolin), flavone glycosides (luteolin-7-*O*-glucoside, apigenin-7-*O*-glucoside), phenolic alcohols (tyrosol, hydroxytyrosol), and secoiridoids (oleacein, oleocanthal, oleuropein, p-HPEA-EA). The predominant phenolic compound found in olive oil is made up by oleuropein and its hydrolytic breakdown products, hydroxytyrosol and tyrosol [23]. Their contribution to human health is very wide: they exert efficacy in prevention and treatment of chronic diseases thanks to their anti-inflammatory, antioxidant, neuroprotective, and immunomodulatory activities [2]. 

In 2011, the European Food Safety Authority (EFSA) published a health claim related to polyphenols in olive oil and their possible protection of blood lipids against oxidative stress. Accordingly, the panel established that 5 mg of hydroxytyrosol and its derivatives (e.g., oleuropein complex and tyrosol) in olive oil should be consumed daily in the context of a balanced diet [24] for sufficient avoidance of oxidative damage [25]. Although, to the best of our knowledge, there are no studies to prove in direct comparison a difference between native EVOO vs. adulterated or poor-quality oils, we assume that the multiple health benefits attributed to the contained bioactive molecules of EVOO might at least be lost, and in the worst case threaten the health of consumers.

### 4.2. Cardiovascular Diseases/Coronary Heart Diseases

There is an abundancy of literature that provides insights into the mechanisms underlying the prevention of CHD by EVOO [29,30]. The overall dietary pattern within the MedDiet plays an additional role, e.g., high fiber positively affecting the microbiome, and even more the myriads of non-fat micro-components. Therefore, the dietary pattern as well as the diet diversity per se present a healthy and balanced composition with energy, macro- and micronutrients, perfectly fitting to most nutritional guidelines. 

Among the beneficial properties of EVOO, its antioxidant effects have been intensively studied because of the link between oxidative stress and atherosclerotic diseases. Oxidative stress is implicated in the pathogenesis of several risk factors of atherosclerosis including hypertension, diabetes, and metabolic syndrome. Guasch–Ferré et al. reported in an observational study that increases of 10 g/day in EVOO intake were associated with a 10% reduction in the risk of cardiovascular events [31]. Several studies suggested that phenolic compounds are important for the cardiovascular benefits of EVOO and showed an improvement of antioxidant capacity [32,33] and a reduction of all markers of oxidative stress [34,35]. A clinical trial on healthy volunteers analyzed the effect on inflammation and oxidative status after administration of 25 g of phenol-rich EVOO with high content of hydroxyphenylethanol [36]. The results highlighted a significant reduction of oxidized LDL, malondialdehyde, triglycerides, and visceral adiposity index. Furthermore, the expression of inflammation and oxidative stress-related genes, as superoxide dismutase 1, upstream transcription factor 1, and catalase was significantly upregulated. 

Moreover, EVOO supplemented to a Mediterranean lunch is capable of blunting oxidative stress by regulating platelet oxidative stress and endothelial dysfunction, as demonstrated by a reduction in Nox2 activation and soluble E-selectin/VCAM1 release, respectively [35]. In humans, the role of EVOO as an anti-atherosclerotic nutrient is also supported by its ability to modulate expression of atherosclerosis-related genes in which LDL oxidation is involved. The anti-inflammatory effect in the vascular wall may be another relevant mechanism, linking EVOO and modulation of cardiovascular events. 

Consumption of a Mediterranean-style diet supplemented with EVOO also in patients with the metabolic syndrome was associated with a significant reduction of systemic vascular inflammation markers (IL-6, IL-7, IL-aa18, and hs-CRP) [37]. Long- and short-term studies demonstrated that EVOO supplementation was also associated with a significant decrease in inflammatory markers, namely Thromboxane-B2 (TXB2) and Leukotriene-B4, which confirmed the anti-thrombotic and anti-inflammatory effects of EVOO in a postprandial state [32]. Studies on subjects at high cardiovascular risk showed that after EVOO supplementation, the blood pressure values, both systolic and diastolic, are reduced [38,39]. In summary, the data demonstrate that EVOO intake is associated with a beneficial impact on CVDs. Intervention studies are consistent with these beneficial effects, as supported by the ability of EVOO to prevent or reduce the inflammatory processes associated with chronic-degenerative diseases, such as cardiovascular-cerebral diseases and cancer.

### 4.3. Metabolic Syndrome with Lipid and Glycemic Control

Positive results were also recorded in a two-year study on subjects with metabolic syndrome [37], showing a decrease in blood pressure values only in women with moderate hypertension supplemented with EVOO and not in the control low-fat diet group [39].

Hypercholesterolemia is another risk factor for CVD. MedDiet has been associated with a reduction in atherogenic cholesterol LDL-C and non-high-density lipoprotein cholesterol (non-HDL-C) levels [40].

Hernáez et al. showed that a one-year intervention with a MedDiet, especially when enriched with virgin olive oil, improves several HDL functions such as cholesterol efflux capacity, cholesterol metabolism, anti-oxidant/anti-inflammatory properties, and vasodilatory capacity in individuals at high cardiovascular risk [41]. Similarly, Covas et al. found that after consuming phenolic olive oils, HDL-C increases, whereas the TC/HDL-C ratio decreases with a concomitant decrease in LDL-C/HDL-C ratio and triglycerides [42].

Nutritional interventions with EVOO may have a positive effect on plasma glucose and lipids. Changes of glycaemic and lipid profiles might have an effect on cardiovascular disease by promoting or aggravating the atherosclerotic process. MedDiet supplemented with EVOO has a beneficial effect on the postprandial metabolic profile by decreasing blood glucose, LDL-C, and ox-LDL and increasing insulin levels in healthy subjects [43]. Furthermore, 10 g of EVOO supplemented to a Mediterranean lunch are able to improve post-prandial glycaemic profiles in patients with impaired fasting glucose. The mechanisms accounting for the positive effect of EVOO are related to incretins up-regulation, as EVOO reduces dipeptidyl peptidase-4 activity with a consequent increase in glucagon-like peptide-1 concentration, which regulates postprandial glycaemia by up-regulating insulin secretion [44]. 

Changes in glycaemic and lipid profiles were observed also in patients with the metabolic syndrome. After 2 years, patients in the intervention group (MedDiet supplemented with EVOO) had significant decreases in the levels of glucose, insulin, total cholesterol, and triglycerides, and a significant increase in the levels of HDL, all of which were greater than the control group [37]. 

Polyphenols might affect glucose metabolism via an inhibition of carbohydrate digestion and absorption, a reduction of glucose release from the liver, or a stimulation of glucose uptake in peripheral tissues. With their antioxidative properties, they might diminish the production of advanced glycosylated end products, such as HbA1cAs, be a potential mechanism of action, and lead to reductions in the glycemic load (especially when replacing carbohydrates with MUFA). The consecutive attenuation in insulin secretion, as well as increased insulin sensitivity, may explain the beneficial effects of MUFA on glycemic control [45].

Summing up, recent data suggest that a Mediterranean-style diet is a possible strategy for the treatment of metabolic syndrome and the reduction of the associated cardiovascular risk. 

Moreover, EVOO supplementation provides an improvement of the post-prandial glucose and lipid profile, and this effect represents an important additional mechanism supporting the role of EVOO as an anti-atherosclerotic nutrient [43]. 

### 4.4. Diabetes Mellitus Type 2

Recent meta-analyses of randomized controlled trials reported beneficial effects on metabolic risk factors in T2DM patients after replacing carbohydrates (~5–10% of total energy intake) in general with MUFA [46,47,48]. The clearest evidence that EVOO may prevent T2DM can be found in the PREDIMED study in which glucose metabolism improved and body weight decreased in 80 cases of new-onset diabetes allocated to the MedDiet + EVOO [3,49,50]. Moreover, specific components of EVOO can also be considered as novel candidates for improving the glycaemic profile in patients with diabetes mellitus. In an interventional study, authors provide evidence that oleuropein lowers postprandial glycaemia by reduction of Nox2 activity in healthy subjects [51].

### 4.5. Cancer

EVOO seems to have a protective role against cancer. For instance, oncology researchers have discovered that oleic acid suppresses the over-expression of the oncogene HER2, which is critical to the etiology, invasion, progression, and metastasis especially of human mammary carcinoma [52]. A recent trial investigated whether hydroxytyrosol improves the antitumor response of women with breast cancer undergoing neoadjuvant chemotherapy, influencing plasma levels of molecules involved in cell proliferation, apoptosis, and metastasis (e.g., tissue inhibitor of metalloproteinases (TIMP-1). Data showed that in women receiving a dietary supplement with 15 mg/day of hydroxytyrosol combined with a specific chemotherapy treatment, the plasma levels of TIMP-1 decreases [53]. 

In vitro studies examined the protective effect of bioactive compounds of oleic acid against neoplastic diseases. One trial demonstrated, for the first time, that olive oil minor components (hydroxytyrosol, oleuropein, pinoresinol, squalene, and maslinic acid) are capable of reversing the proliferative effect induced by oleic acid on intestinal epithelial cell cultures [54]. Another recent trial evaluated the in vitro anticancer and chemopreventive potential of two EVOO extracts (tyrosol and hydroxytyrosol) and secoiridoid derivatives (oleocanthal and oleacein) on cutaneous non-melanoma skin cancer models. Results demonstrated that phenolic EVOO extracts can block molecular steps that occur after the initial UV radiation exposure and before or during tumor development. In particular, these findings showed that secoiridoid derivatives contribute more than simple phenols to the mechanism of action of EVOO extracts [55].

A systematic review and a metanalysis of 19 case-control studies, including 23,340 controls and 13,800 patients, concluded that high EVOO intake has a protective role on the risk of breast cancer (logOR = −0.45 95%CI −0.78 to −0.12), and digestive cancer (logOR = −0.36, 95%CI −0.50 to −0.21) compared with the lowest intake [56]. 

Epidemiological studies have revealed that women who follow a MedDiet have lower rates of breast cancer than those who did not [57,58]. EVOO has been shown to contribute to the prevention of a dangerous form of skin cancer, malignant melanoma [59]. With regards to melanoma, the antioxidant properties of EVOO help to counter oxidation by the sun. Researchers have established the oleocanthal ability to provide potent anti-inflammatory properties, which inhibit COX enzymes in the same way as the non-steroidal anti-inflammatory drug (NSAID) ibuprofen [60]. For the first time, a study revealed that oleocanthal may provide anti-prolific activity against melanoma cells by inhibiting melanoma cell growth in a concentration-dependent manner, that is, the more the cells are exposed to oleocanthal, the less they proliferate and become cancer causing. 

### 4.6. Weight Control

Chronic obesity is a situation of chronic systemic inflammation [61] and can contribute, for instance, to the development and/or severity of asthmatic and probably also allergic diseases [62]. Weight control, therefore, is of uttermost importance, and MedDiet could contribute also here. However, there is limited, but suggestive, evidence for the effects on the body weight regarding the reduction of fat mass with consequent increases of muscle mass. A trial evaluated the effects of hydroxytyrosol on the expression of genes and microRNAs involved in TNF-α and macrophage-induced inflammatory and dysmetabolic phenotype of human adipocytes. Results demonstrated that hydroxytyrosol can modulate the adipocyte gene expression profile through mechanisms involving a reduction of oxidative stress and NF-κB inhibition and may blunt macrophage recruitment, preventing the deregulation of pathways involved in obesity-related diseases [63]. Further dietary intervention studies in humans are needed [64].

### 4.7. Allergic Diseases

Recent systematic reviews and meta-analysis suggest that the Mediterranean dietary pattern is inversely related to asthma symptoms in children; in particular, when considering all studies together, data showed that a MedDiet is a protective factor for “current wheeze”, and “asthma ever” but not for “severe current wheeze” [65,66].

A possible explanation for the beneficial/prophylactic effects regarding asthmatic symptoms is the richness in bioactive compounds that may prevent or limit inflammatory responses in the airways by reducing reactive oxygen species and inhibiting lipid peroxidation.

If we consider olive oil specifically, only one trial is available in the last ten years: it’s a population multi-case study of Italian adults, where authors investigated the association of dietary fatty acids and of olive oil with asthma and rhinitis [67]. Results showed that intakes of MUFA and oleic acid were associated with a reduced risk of current asthma (CA), even when they were categorized in quartiles; the subjects in the highest quartile of oleic acid uptake had less than half the risk of having CA compared to those in the lowest quartile. Moreover, authors found that, when considering olive oil intake as a continuous variable, the risk of being a case of CA decreased by 20% for an increase of 10 g/day in olive oil intake.

Another trial investigated the specific role of the phenolic compound hydroxytyrosol (HT) that exerts notable anti-inflammatory effects [68]. The authors studied the ability of HT to increase a Parietaria allergen-induced IL-10 secretion ex vivo in PBMCs from healthy volunteers. This pilot work demonstrated that the co-administration of HT and an allergenic molecule to PBMCs can potentiate a tolerogenic immune response via an increase of IL-10 secretion, suggesting the capability of HT to potentiate a healthy immune response.

Importantly, to confirm the efficacy of a MedDiet in preventing immune-mediated allergic diseases, the worldwide prevalence numbers for allergic rhinoconjunctivitis and asthma underline a possible association, with lower numbers especially in the Eastern Mediterranean, while higher numbers are observed, for instance, in English language countries [69]. 

EVOO polyphenols can also be used to manage skin inflammatory conditions. In the study by Aparicio–Soto et al., results demonstrated that EVOO phenols exert anti-inflammatory effects on human keratinocytes suppressing key epidermal cytokines [70].

On the whole, convincing epidemiological and experimental literature indicates that olive products may play a key role in the health benefits of the MedDiet, and a model for healthy eating, contributing to a favorable health status and therefore a better quality of life.

## 5. Summary

Does [only] olive oil matter? For sure, olive oil being the main source of fat in a Mediterranean diet (Figure 1) has direct (tocopherols, polyphenols, mono-unsaturated FA) and indirect (lower saturated fats, balanced linoleic/alpha linolenic acid) effects on the immune system and inflammatory responses (Figure 2). 

The overall dietary pattern within the MedDiet plays an additional role, e.g., high fiber positively affecting the microbiome, and even more the myriads of non-fat micro-components. Therefore, the dietary pattern as well as the diet diversity per se present a healthy and balanced composition with energy, macro- and micronutrients perfectly fitting to most nutritional guidelines. In addition, environmental, socio-cultural and life-style factors seem to present an uttermost important additional precondition for this diet to be effective in immunomodulation. The food being grown locally and sustainably, being minimally processed and eaten freshly, prepared at home, eaten in a pleasurable familiar conviviality, the moderate consumption of wine, as well as people having a rest after the meal, together with regular physical activity—all these factors seem to be necessary equipment for the MedDiet to positively affect human health. Furthermore, the fact that Mediterranean populations live in environments connected to the sea, coasts, and mountains, allowing them to exercise in everyday life within their millenarian historical evolution, should not be underestimated. Therefore, all these factors need to be considered when performing transferability studies of this diet to people all over the world.

## Figures and Tables

**Figure 1 nutrients-11-02941-f001:**
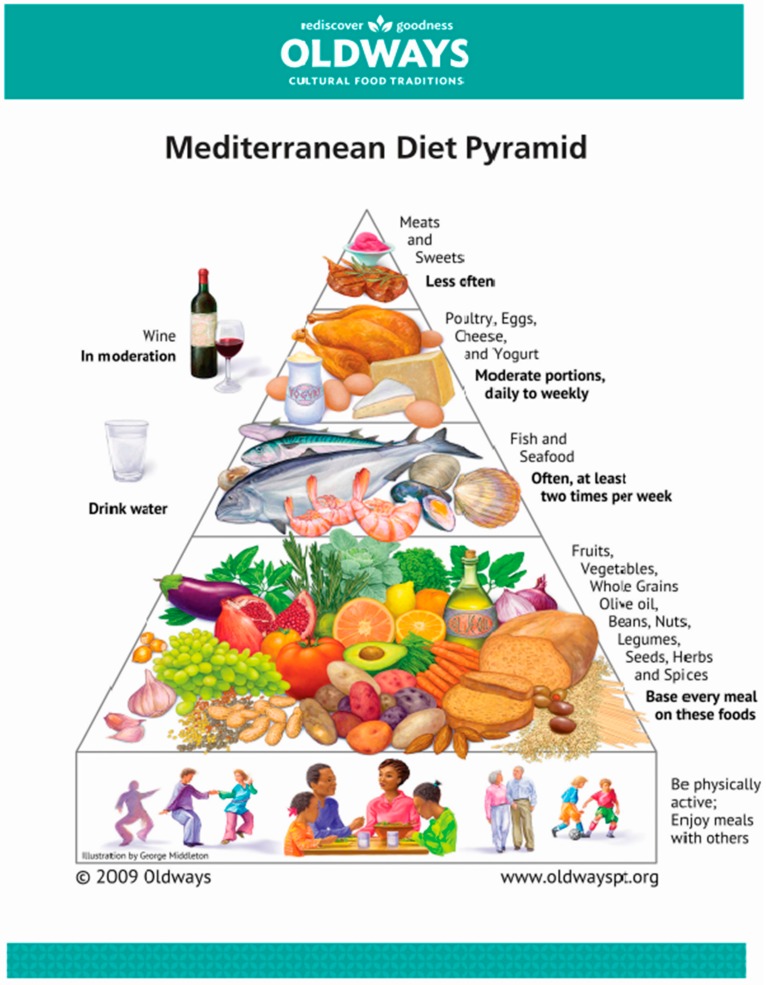
The Mediterranean Diet Pyramid with olive oil as an important nutritional fat source. Image credit: Copyright by Oldways and [9], used with kind permission.

**Figure 2 nutrients-11-02941-f002:**
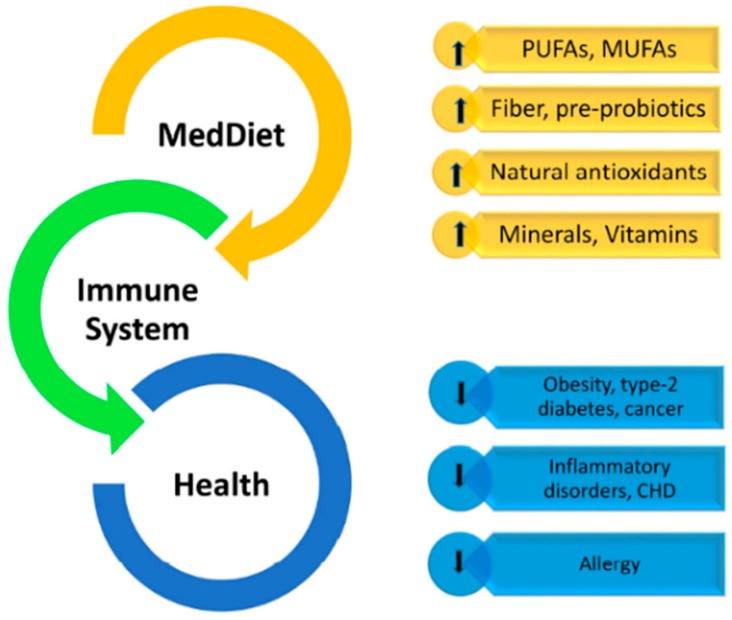
The multistep influence of the Mediterranean diet (MedDiet) on the immune system and related disorders.

**Table 1 nutrients-11-02941-t001:** Nutrients in a Mediterranean diet compared to the guidelines for daily nutrient uptake.

Nutrients	Distribution in Mediterranean Diet	Recommendations for 19–50-Year Old Americans *
Total fat (% of total calories)	25–35	20–35
Saturated fat (% of total calories)	≤8	≤10
Cholesterol (mg)	≤300	Not restricted
Sodium (mg)	≤2300	≤2300
Potassium (mg)	≥4700	≥4700
Carbohydrates (% of total calories)	45–65	45–65
Protein (% of total calories)	10–35	10–35
Vitamin A (µg)	≥900	women: 700; men: 900
Vitamin C (µg)	≥90	women: 75; men: 90
Calcium (mg)	≥1200	≥1000
Iron (mg)	≥18	women: 18; men: 8

Table adapted from [10] (with kind permission). * Recommendations according to the U.S. Department of Health and Human Services and the U.S. Department of Agriculture. 2015–2020 Dietary Guidelines for Americans. 8th Edition. December 2015. Available at [11].

**Table 2 nutrients-11-02941-t002:** Nutritional components present in extra virgin olive oil.

Component	Amount Per 100 g Olive Oil
Energy	884 kcal/3699 kJ
Carbohydrates, fiber	0–0.2 g
Protein	0
Fat saturated FA mono-unsaturated FA poly-unsaturated FA	100 g 14 g 73 g (up to 73% of RDA) 13 g
Cholesterol	0
Vitamin A	0–157 µg
Vitamin E	0–37 mg (up to 72–96% RDA)
Vitamin K	55–60 µg (up to 50–75% RDA)
Sodium	1–2 mg
Potassium	0–1 mg
Calcium	0–1 mg
Magnesium	0–1 mg
Phosphor	0–2 mg
Iron	100–560 µg (up to 7% RDA)
Zinc	10–60 µg
Copper	0–70 µg
Chlorophyll	0.5–1.6 mg
(Poly)phenols and phenolic compounds (*n* = 36), e.g., Oleuropein Tyrosol Hydroxytyrosol Oleocanthal	28–221 mg

Values might differ considerably according to olive cultivar, climate conditions, and production process of oil [24]. Data taken and combined from [26,27,28].

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
