# Peer review of "The Secrets of the Mediterranean Diet. Does [Only] Olive Oil Matter?"

_nutrients, 2019, doi:10.3390/nu11122941_

Round 1

Reviewer 1 Report

I have added my edits to the PDF. Many sentences needs to be reworded. 

Author Response

Point-by-Point-Reply

Reviewer 1

I have added my edits to the PDF. Many sentences needs to be reworded.

Line 22: Not defined.

RESPONSE: Thanks, we exchanged “FA” by “fatty acids”.

Line 46: Vitamins and trace elements are also involved in the maintenance of the immune response. The Mediterranean dietary pattern is not only exemplified as a prudent choice of lifestyle....protects against major chronic diseases.

RESPONSE: Thanks, we changed the sentence as you suggested.

Line 79: Figgure > Figure

RESPONSE: Thanks, we corrected the error.

Line 86: Table 1 refers to dietary nutrients and not nutrient uptake. Please reword.

RESPONSE: We replaced the referral to Table 1 within the sentence to “nutrient content” and reworded to “…it was shown that the content in MedDiet was nearly totally according to the actual guidelines for daily nutrient uptake…”

Line 89: Delete

RESPONSE: We removed the word.

Line 90: Not defined.

RESPONSE: Thanks, we added the term “fatty acids”

Line 93: Delete

RESPONSE: We removed the words.

Line 94: Delete

RESPONSE: Deleted.

Line 106: Delete

RESPONSE: Deleted.

Line 109: Reword sentence

RESPONSE: We reworded the sentence to “Such an independent measure based on desirable dietary patterns is the Mediterranean Diet Score (11), suggested by Trichopoulou et al. in 2003 (12).” and hope that the reviewer can agree on that.

Line 125: It is not clear how diet diversity is scored according to the previously mentioned section. 

RESPONSE: We deleted the part referring to diet diversity scoring here in this sentence, and hope that the explanation with exact point scoring given in line 110-122 is clearly formulated.

Line 127: Change to bioactive

RESPONSE: Thanks, we changed the word “Responsible” to “Bioactive”

Lines 128-130:  Remove/ This part of the sentence can be removed.

RESPONSE: We changed the sentence as suggested.

Line 132: Repetition, please remove.

RESPONSE: Thanks for pointing this out, we removed the repetition.

Line 133: Reword

RESPONDE: We changed the sentence to “Traditionally, the high content in MUFAs was considered to be responsible for the protective effects of EVOO.”

Line 135: Fatty acids

RESPONSE: Thanks, we added the word “fatty”

Line 148: (Table 1): Realign

RESPONSE: Done.

Line 149 (Table 1): Need references

RESPONSE: References included now in footnote to table.

Lines 162-165: Need references

RESPONSE: References given now:

[23] Bendinelli, B.; Masala, G.; Saieva, C.; Salvini, S.; Calonico, C.; Sacerdote, C.; Agnoli, C.; Grioni, S.; Frasca, G.; Mattiello, A.; et al. Fruit, vegetables, and olive oil and risk of coronary heart disease in Italian women: The EPICOR Study. Am. J. Clin. Nutr. 2011, 93, 275–283.

[24] Review: Annalisa Romani 1,*, Francesca Ieri 1 , Silvia Urciuoli 1 , Annalisa Noce 2,* , Giulia Marrone 2,3 , Chiara Nediani 4 and Roberta Bernini 5. Health Effects of Phenolic Compounds Found in Extra-Virgin Olive Oil, By-Products, and Leaf of Olea europaea L. Nutrients 2019, 11, 1776; doi:10.3390/nu11081776 (in press).

Line 173: Remove

RESPONSE: Thanks, we removed the word.      

Line 176: Remove

RESPONSE: Thanks, we removed the phrase.

Line 185: There is no data reported in this paper. Change to ....the data demonstrates that EVOO intake is associated...

RESPONSE: Thanks, we changed the sentence as you suggested.

Line 186: Intervention

RESPONSE: Thanks, we changed the word “Interventional” to “Intervention”.

Line 195: ..showed that a one year intervention with...

RESPONSE: Thanks, we changed the sentence as suggested.

Line 201: Reword

RESPONSE: We reworded the sentence to “Nutritional interventions with EVOO may have a positive effect on plasma glucose and lipids”.

Lines 214-220: Need references for all these statements

RESPONSE:  References given now:

[39] Schwingshackl L, Lampousi AM, Portillo MP, Romaguera D, Hoffmann G5, Boeing H. Olive oil in the prevention and management of type 2 diabetes mellitus: a systematic review and meta-analysis of cohort studies and intervention trials. Nutr Diabetes. 2017 Apr 10;7(4):e262.

[37] Violi F, Loffredo L, Pignatelli P, Angelico F, Bartimoccia S, Nocella C, Cangemi R, Petruccioli A, Monticolo R, Pastori D, Carnevale R. Extra virgin olive oil use is associated with improved post-prandial blood glucose and LDL cholesterol in healthy subjects. Nutr Diabetes. 2015 Jul 20;5:e172.

Line 230: change word

RESPONSE: Thanks, we changed the phrase to “80 cases of new-onset diabetes” as used in the vited paper.

Line 234: I think this only refers to breast cancer.

RESPONSE: The authors in the cited article state that ”…can suppress the overexpression of HER2 (erbB-2), … in several human cancers”. However, we rephraised now to “For instance, oncology researchers have discovered that oleic acid suppresses the over-expression of the oncogene HER2, which is critical to the etiology, invasion, progression and metastasis especially of human mammary carcinoma”.

Line 256: Remove

RESPONSE: We removed the sentence.

Line 258: remove

RESPONSE: Removed.

Line 258: suggest

RESPONSE: Thanks, we changed “suggesting” to “suggest”.

Line 283: EVOO polyphenols can also be used to manage skin inflammatory conditions.

RESPONSE: Thanks, we changed the sentence as you suggested.

Reviewer 2 Report

The manuscript entitled: The secrets of Mediterranean diet. Does [only] olive oil 2 matter? is an interesting review I have learned a lot. This is despite I feel the title is somewhat misleading. I was hoping to finish the reading with a clearer picture as to what is olive oil and what is not. Therefore it feels that with a little more work the reader could get a clearer picture. In addition, at places the sentences are very confused, maybe a further reading would help to make sure that the sentences are what they were intended to be. For Example:

1) As a common central feature have the usage of olive oil as the main source of fat.

2)The main dietary sources, already mentioned, responsible for these valuable compounds and mentioned above are olive oil, wine, fish and vegetables, with high consumption of tomatoes, onions, garlic and herbs like oregano, mint, rosemary, parsley and dill (8)

3) Traditionally, the high content in MUFAs was considered the main responsible for the protective effects. Indeed, the concentration of oleic acid (C18:1 n-9) is much higher (55-83%) than that of the other acids (linoleic, palmitic, or stearic acids), ranging between 3% and 21%.

4) In 2011, the European Food Safety Authority (EFSA) published a health claim related to
polyphenols in olive oil and their possible beneficial physiological effects. The Panel established a cause-and-effect relationship between the consumption of olive oil polyphenols (standardised by their content of hydroxytyrosol and its derivatives) and protection of LDL particles from oxidative damage.

These were a few examples where clarifying the sentences will significantly improve the meaning.

while this is not a deal-breaker, it would have been interesting to hear a few sentences about food fraud in olive oils and how this could affect the efficacy of olive oil.

small issue, define the abbreviation where they are mentioned first.

Author Response

Point-by-Point-Reply

Reviewer 2

The manuscript entitled: The secrets of Mediterranean diet. Does [only] olive oil 2 matter? is an interesting review I have learned a lot. This is despite I feel the title is somewhat misleading. I was hoping to finish the reading with a clearer picture as to what is olive oil and what is not. Therefore it feels that with a little more work the reader could get a clearer picture.

RESPONSE: We thank the reviewer for this constructive criticism and in accordance with suggestion of reviewer 3 we tried to put more emphasis on the role of EVOO and its active components in their role of disease protection (e.g. references 30, 45, 47, 48, 49 and 57).

In addition, at places the sentences are very confused, maybe a further reading would help to make sure that the sentences are what they were intended to be. For Example:

1) As a common central feature have the usage of olive oil as the main source of fat.

RESPONSE: We rephrased to “Their common central feature is the usage of olive oil as the main source of fat”.

2) The main dietary sources, already mentioned, responsible for these valuable compounds and mentioned above are olive oil, wine, fish and vegetables, with high consumption of tomatoes, onions, garlic and herbs like oregano, mint, rosemary, parsley and dill (8)

RESPONSE: We rephrased to “The main dietary sources providing these valuable compounds are olive oil, wine, fish and vegetables, especially tomatoes, onions, garlic and herbs like oregano, mint, rosemary, parsley and dill (8).”

3) Traditionally, the high content in MUFAs was considered the main responsible for the protective effects. Indeed, the concentration of oleic acid (C18:1 n-9) is much higher (55-83%) than that of the other acids (linoleic, palmitic, or stearic acids), ranging between 3% and 21%.

RESPONSE: We rephrased to “Traditionally, the high content in MUFAs was considered to be responsible for the protective effects of EVOO . Indeed, the concentration of the MUFA oleic acid (C18:1 n-9) is much higher (55-83%) than that of the other FA (linoleic, palmitic, or stearic acids), which range between 3% and 21%.”

4) In 2011, the European Food Safety Authority (EFSA) published a health claim related to polyphenols in olive oil and their possible beneficial physiological effects. The Panel established a cause-and-effect relationship between the consumption of olive oil polyphenols (standardised by their content of hydroxytyrosol and its derivatives) and protection of LDL particles from oxidative damage.

RESPONSE: We rephrased to “In 2011, the European Food Safety Authority (EFSA) published a health claim related to polyphenols in olive oil and their possible protection of blood lipids against oxidative stress. Accordingly, the panel established that 5 mg of hydroxytyrosol and its derivatives (e.g. oleuropein complex and tyrosol) in olive oil should be consumed daily in the context of a balanced diet (21) for sufficient avoidance of oxidative damage.”

These were a few examples where clarifying the sentences will significantly improve the meaning.

RESPONSE: Thank you for pointin them out, we tried to improve them as much as possible.

while this is not a deal-breaker, it would have been interesting to hear a few sentences about food fraud in olive oils and how this could affect the efficacy of olive oil.

RESPONSE: We thank the reviewer for this remark and the question is quite interesting; however, while there is a bunch of literature describing the techniques to detect frauds in olive oil (e.g. Carranco et al, Authentication and Quantitation of Fraud in Extra Virgin Olive Oils Based on HPLC-UV Fingerprinting and Multivariate Calibration. Foods. 2018 Mar 21;7(4). pii: E44. doi: 10.3390/foods7040044), we were not able to find any literature on direct comparison of the effect of native versus adulterated olive oil, and therefore put in that adulteration and production low-quality oil can also affect the health benefits as assumption into the text: “Although to the best of our knowledge there are no studies to prove in direct comparison a difference between native EVOO vs. adulterated or poor-quality oils, we assume that the multiple health benefits attributed to the contained bioactive molecules of EVOO might at least be lost, and worst case threaten the health of consumer.”

small issue, define the abbreviation where they are mentioned first.

RESPONSE: We tried to define all abbreviations at first site mentioned.

Reviewer 3 Report

This review would like to stess on the important of olive oils in diet foods/ diest plans. However, I am afraid that I could not find any significant reports and/or evidences related to the effects of olive oils complemented with other foods except for the references that have been reported so fa as similar and same results.  If this review would be meaningful, I think, it should contains strong/definite evidence that the extract olive oils (as the authors clalimed) can have or enfluence the dietary effects, however, most of the results shown in this reveiw are already found and proved... Can the authors could cleary provide the single effects of olive oils in taking all the foods same times?

Author Response

Point-By-Point-Reply 

Reviewer 3

This review would like to stess on the important of olive oils in diet foods/ diest plans. However, I am afraid that I could not find any significant reports and/or evidences related to the effects of olive oils complemented with other foods except for the references that have been reported so fa as similar and same results.  If this review would be meaningful, I think, it should contains strong/definite evidence that the extract olive oils (as the authors clalimed) can have or enfluence the dietary effects, however, most of the results shown in this reveiw are already found and proved... Can the authors could cleary provide the single effects of olive oils in taking all the foods same times?

RESPONSE:

We thank the referee for this input. We revised the whole manuscript to stress the explicit effect of olive oil and its components, and added recent literature concerning the health benefits from extract olive oils. Here the new references:

[30] Perrone MA, Gualtieri P, Gratteri S, Ali W, Sergi D, Muscoli S, Cammarano A, Bernardini S, Di Renzo L, Romeo F. Effects of postprandial hydroxytyrosol and derivates on oxidation of LDL, cardiometabolic state and gene expression: a nutrigenomic approach for cardiovascular prevention. J Cardiovasc Med (Hagerstown). 2019 Jul;20(7):419-426.

[45] Carnevale R, Silvestri R, Loffredo L, Novo M, Cammisotto V, Castellani V, Bartimoccia S, Nocella C, Violi F. Oleuropein, a component of extra virgin olive oil, lowers postprandial glycaemia in healthy subjects. Br J Clin Pharmacol. 2018 Jul;84(7):1566-1574.

[47] Ramirez-Tortosa C, Sanchez A, Perez-Ramirez C, Quiles JL, Robles-Almazan M, Pulido-Moran M, Sanchez-Rovira P, Ramirez-Tortosa M. Hydroxytyrosol Supplementation Modifies Plasma Levels of Tissue Inhibitor of Metallopeptidase 1 in Women with Breast Cancer. Antioxidants (Basel). 2019 Sep 11;8(9).

[48] Storniolo CE, Martínez-Hovelman N, Martínez-Huélamo M, Lamuela-Raventos RM1, Moreno JJ. Extra Virgin Olive Oil Minor Compounds Modulate Mitogenic Action of Oleic Acid on Colon Cancer Cell Line. J Agric Food Chem. 2019 Oct 16;67(41):11420-11427.

[49] Polini B, Digiacomo M, Carpi S, Bertini S, Gado F, Saccomanni G, Macchia M, Nieri P, Manera C, Fogli S. Oleocanthal and oleacein contribute to the in vitro therapeutic potential of extra virgin oil-derived extracts in non-melanoma skin cancer. Toxicol In Vitro. 2018 Oct;52:243-250.

[57] Scoditti E, Carpi S, Massaro M, Pellegrino M, Polini B, Carluccio MA, Wabitsch M, Verri T, Nieri P, De Caterina R. Hydroxytyrosol Modulates Adipocyte Gene and miRNA Expression Under Inflammatory Condition. Nutrients. 2019 Oct 17;11(10).

Round 2

Reviewer 3 Report

 in my opinion, it is very difficult to have such a solid conclusion in human clinical experiments even though I think they did not (?) have carried out this kinds of experiments, only the results from references and random samples in this region (even not Metasheet analysis)